# Post-COVID-19 Syndrome in Outpatients and Its Association with Viral Load

**DOI:** 10.3390/ijerph192215145

**Published:** 2022-11-17

**Authors:** Daniel Alberto Girón Pérez, Aimee Fonseca-Agüero, Gladys Alejandra Toledo-Ibarra, Jaqueline de Jesus Gomez-Valdivia, Karina Janice Guadaluope Díaz-Resendiz, Alma Benitez Benitez-Trinidad, Francisco Fabian Razura-Carmona, Migdalia Sarahy Navidad-Murrieta, Carlos Eduardo Covantes-Rosales, Manuel Ivan Giron-Pérez

**Affiliations:** 1Laboratorio Nacional de Investigación Para la Inocuidad Alimentaria (LANIIA) Unidad Nayarit, Universidad Autónoma de Nayarit, Tepic 63000, Mexico; 2Laboratorio de Psicofisiología y Conducta, Unidad Académica de Ciencias Sociales, Universidad Autónoma de Nayarit, Tepic 63000, Mexico

**Keywords:** post-COVID-19 syndrome, perception, mental health, viral load, SARS-CoV-2

## Abstract

Introduction: The COVID-19 pandemic is the result of the SARS-CoV-2 virus, which has caused more than 100 million infections and more than 2.5 million deaths worldwide, representing a serious public health problem. The gold method for detecting this virus is qRT-PCR, which is a semiquantitative technique where the viral load can be established through its cycle threshold (Ct). It has also been reported that COVID-19 generates long-term symptoms (post-COVID-19). Methods: After three months, a survey was performed on 70 COVID-19 confirmed patients; subsequently, we divided them into four groups (persistent symptoms, chemo-sensitive, cognitive issues, and changes in habit) in order to determine the correlation between viral load and post-COVID-19 symptoms. Results: Data show that fatigue, nervousness, anosmia, and diet changes are common long-term symptoms; in addition, a negative correlation was found between viral load and the number of post-COVID-19 symptoms. Conclusion: COVID-19 generates long-term symptoms which can cause problems with psychological and social repercussions.

## 1. Introduction

The severe acute respiratory syndrome coronavirus 2 (SARS-CoV-2) is an enveloped virus belonging to the beta coronaviruses and is the cause of the coronavirus disease 2019 (COVID-19). Also, it is the cause of the pandemic that sweeps the world today, which originated in the Wuhan region of China [1,2,3]. This disease has generated more than 2.5 million deaths and over 100 million infected people around the world, causing a serious public health problem [2].

SARS-CoV-2 detection occurs through the quantitative real-time-polymerase reaction chain (qRT-PCR) technique, amplifying the E gene (envelope); in addition, the viral load (number of copies) can be associated with the cycle threshold (Ct) and can therefore monitor the development of COVID-19 [4].

The association between Ct value and viral load (based on E or N gene) has been reported in different works [5,6], where it has been established that at a low Ct value there is a high viral load, while the opposite is the case when there is a high value of this parameter. It has also been reported that viral load has repercussions on the number of symptoms and signs that may develop after viral infection [7]. Post-COVID-19 symptoms can manifest as cognitive and psychological disturbances, such as confusion, depression, memory loss, insomnia, and in some cases mania or psychosis [8]; in addition, persistent symptomatology can occur, such as headaches, cough, partial or total loss of smell/taste, which can last for weeks or months [9].

COVID-19 infection not only has an impact on the symptoms that occur during the disease (headache, anosmia, diarrhea, ageusia, shortness of breath) but also causes changes after the disease, among which we find alterations in habits, behavior, diet (development of eating problems), or sleep (sleep disturbance), as well as an impact on the work and social environment [10]. This phenomenon is called “post-COVID-19 syndrome” which poses a new challenge for global public health.

Post-COVID-19 syndrome has had different pathologies associated with pulmonary (hypoxia, lung damage, or fibrosis), hematologic (thromboembolic events, longer duration of inflammatory events), cardiovascular (dyspnea and prolonged chest pain), neurological (sleep disruption, depression, fatigue), renal (kidney damage), endocrine (bone demineralization, uncontrolled diabetes, thyroiditis), and dermatologic (hair loss) systems [11]. All these conditions should be approached from a multidisciplinary point of view to identify variables that can predict the susceptibility of these conditions. One of these variables is the viral load, which can be defined as the amount of a virus in an organism (bloodstream or tissues), typically quantified by virus particles per milliliter [12].

The relationship between SARS-CoV-2 virus detection at different points in time and its viral load may aid in the clinical interpretation of post-COVID-19 syndrome. In addition, it has been stated that the virus can survive longer in feces compared to respiratory tract secretions [13]. This may result in viral proteins that may not cause infection, but trigger and exacerbate a prolonged immune response, which could explain the presiding symptoms associated with COVID-19 [14].

There are few studies regarding whether the viral load determined by Ct has any impact on the different post-COVID-19 effects, either in persistent symptomatology or whether these alterations are due to risk factors associated with the individual environment of patients [11]. Therefore, this paper aims to evaluate the post-COVID-19 symptoms prevalent in patients and to analyze the correlation between Ct and these symptoms after infection.

## 2. Materials and Methods

### 2.1. Patients

We included 70 COVID-19 cases, confirmed by a positive result in the real-time reverse transcription-polymerase chain reaction test (RT-PCR) (after three months the survey was performed). For swab sample collection, subjects were asked to avoid eating food, drinking water, and brushing their teeth, at least 4 h before sample collection. All cases were outpatients of Tepic, Nayarit.

### 2.2. Ethical Aspects

Postulates of the Declaration of Helsinki have been carried out and were approved by the local bioethics panel (“Comisión Estatal de Bioética del Estado de Nayarit”) (registry number CEBN/03/20). All patients who fulfilled the inclusion criteria were informed of the purpose of the study. All of them expressed verbal consent and signed the informed consent before sample collection, clinical data, and demographic information were collected.

### 2.3. Determination of SARS-CoV-2 in Patients

The molecular protocol was validated by the Mexican Ministry of Health (SSA-Mexico, Cd de Mexico). In brief, each patient was swabbed with nasal and oropharyngeal samples, which were subjected to RNA extraction by QIAmp Viral RNA Mini Kit (Qiagen, Cat No./ID: 1020953, Germantown, MD, USA). The RT-PCR test was performed using the StarQ One-Step RT-qPCR Kit (Qiagen, Cat No./ID: 210210, Germantown, MD, USA). SARS-CoV-2 detection was performed using primers and probes to detect the viral E gene (E_Sarbeco_F: ACAGGTACGTTAATAGTTAATAGCGT, E_Sarbeco_R: ATATTGCAGCAGCAGTACACGC-ACACACA, E_Sarbeco_P1: FAM-CACTAGCCATCCTCCTTACTGCGCTTCG-BBQ) as well as a human cell control (RNase P gene). (RNAseP F: AGATTTTTGGACCTCTGGAGCG, RNAseP R, GAGCGGCTGCTCCACAAGT, RNAseP P1, FAM-TTCTGACCTGAAGGCTGCGG-BHQ1) [15]. The SARS-CoV-2 detection was performed using primers, probes, and PCR conditions according to the Berlin protocol and validated by InDRE (Instituto de Diagnóstico y Referencia Epidemiológica). All samples were analyzed in the ABI Prism 7500 Sequence Detector System (Applied Biosystems, Waltham, MA, USA).

### 2.4. Survey

The questionnaire was administered to 76 patients by the psychology department of the Universidad Autónoma de Nayarit (UAN) and the questions were open and confidential.

### 2.5. Data Analysis

All statistical analysis was carried out using Gradpadh 5.0 (London, UK) using *t*-tests (*p* < 0.05). Descriptive statistics are reported as means with standard deviations (SD), matching patients according to their sex (men and women), and compared with the cycle threshold values. For the study of the symptomatology, a frequency analysis was carried out.

## 3. Results

A total of 76 outpatients (40 men and 36 women) were evaluated, all from Tepic, Nayarit. The patients age range was between 20 to 70 years, the average age in men was 45 years, while in women it was 30 years. The survey was conducted by telephone by the psychology department of the Universidad Autónoma de Nayarit (UAN). The patients surveyed had been negative for SARS-CoV-2 infection for more than 3 months (determination made by qRT-PCR). 70 of the patients reported mild symptoms during infection (92%), while 6 patients (8%) had no symptoms during their SARS-CoV-2 infection (Table 1).

For post-COVID-19 symptoms analysis, patients were divided into four groups: persistent symptomatology, cognitive skill changes, chemo-sensitive problems, and habit changes. Data show that the most common symptom in the persistent symptomatology group was fatigue 60% (46/70), in the group of chemo-sensitive alterations it was smell alteration 25% (19/70 people), while in the cognitive skill changes group it was increased nervousness 52% (40/70 people), and in the habit group it was a change in eating habits 42% (32/70 people) (Table 1).

Gender can cause sex bias in the presentation of post-COVID-19 symptoms, so an analysis between the main symptoms and sex was performed. In men, the main persistent symptom is fatigue (95%), while in women it is headache (41%); in the section of chemo-sensitive alterations augesia prevailed in men (35%) and anosmia in women (30%) (Figure 1A,B).

Regarding cognitive alterations, the main impairment in men was nervousness (65%), while in women it was forgetfulness (27%). In changes in habits, there was no difference between men and women, since both reported similar percentages of alterations in diet (~55%) and sleep (~30%) (Figure 2A,B). These data suggest that there is a sex bias in the manifestation of post-COVID-19 symptoms.

Viral load is another important parameter to know (Table 2) [16] for whether people can develop post-COVID-19 symptoms. Our results show that, at a low Ct value [15,16,17,18,19,20], patients have between 15 to 20 post-COVID-19 symptoms; likewise, this is reflected when analyzing patients who had an intermediate Ct [21,22,23,24,25,26,27,28,29] who presented between 6 to 11 symptoms, and, finally, there were patients with a low Ct [30,31,32,33,34,35,36,37] who had few or no symptoms (Figure 3A). Therefore, a Pearson correlation was performed between the Ct value and the number of post-COVID-19 symptoms (r = 0. 74 and *p* = 0.0001) (Figure 3B). This indicates that the higher the viral load, the more post-COVID-19 symptoms are likely to occur; however, this may be associated with other risk factors.

## 4. Discussion

Infections by members of the coronavirus family such as severe acute respiratory syndrome coronavirus (SARS) and Middle East respiratory syndrome coronavirus (MERS) also cause post-infection symptoms. Lam and colleagues described that patients who recovered from SARS presented with fatigue and psychiatric disorders (depression, panic, obsessive states, and post-traumatic stress) [17]; in addition, Lee and colleagues showed that healthcare workers who treated patients with MERS presented with sleep disturbances and numbness [18].

The presence of symptoms following SARS-CoV-2 infection is of clinical relevance as it impairs the quality of life and creates a psychosocial risk due to the arrangement of adverse effects that persist over time, referred to as “post-COVID-19 syndrome” [13].

In the present investigation, the main alterations were divided into four groups (persistent symptomatology, cognitive skill changes, chemo-sensitive problems, and habit changes.). The most common persistent symptomatology was fatigue, which manifested itself in more than 50% of the patients surveyed, where 95% of the men presented this affectation, while in women it was 22%. The second most common symptom was headache, where, interestingly, a higher percentage of women (41%) reported this symptom, while only 32% of men did. This is in agreement with what was reported by Townsend et al. in 2020, where they showed that fatigue is the first post-COVID-19 manifestation [19]. In addition, in patients who required hospitalization, it has been observed that headache is a persistent symptom post-SARS-CoV-2 infection [20].

Anosmia and ageusia are symptoms used as biomarkers for SARS-CoV-2 infection. These chemo-sensitive alterations (smell and taste) persist in SARS-CoV-2 negative subjects, indicating that SARS-CoV-2 possibly affects the olfactory and taste nerve endings. These results are consistent with those described by Vaira, where the loss of these senses was detected up to 60 days after infection [21]. However, these effects have not been fully understood and more studies are needed to comprehend entirely the mechanism of action of SARS-CoV-2 infection with regard to the sense of smell and taste [22].

SARS-CoV-2 infection can reach the CNS through the olfactory tract and access the cortex, basal ganglia, and midbrain, which may be affected during propagation [23] supporting the existence of neurological symptoms such as headache, anosmia, dysgeusia, dizziness, and impaired consciousness; thus, olfactory dysfunctions (OD) throughout the pandemic became part of the symptoms that carry a warning sign of a possible ongoing infection, even when they appear in isolation The mechanisms that can cause OD have not yet been fully defined. In this context, studies point to several possibilities that can lead to olfactory impairment, such as conductive loss (due to edema in the olfactory cleft), injury to the respiratory epithelium (due to the local inflammatory response with an increase in pro-inflammatory cytokines and chemokines such as IL-6 and IFN-γ), or lesion in the olfactory bulb, with neuronal damage. Symptoms begin on average within the first week of infection, vary in duration, and can last for weeks or month [24].

COVID-19 infection has been reported to cause long-term neurological alterations, such as cerebrovascular diseases or inflammatory myopathies type 1 [25]. Likewise, other symptoms such as anosmia and augesia can manifest after the end of the infection and can be used as prognostic markers for neuronal diseases. Other manifestations associated with this disease include the increase of migraine or cases of epilepsy; however, the evidence is still unclear because these may occur due to the infection or to the obligatory confinement [23,26,27].

Post-infection symptoms also affect cognitive abilities. It has been found that MERS or SARS-CoV can cause, mania, stress, and psychosis as well as changes in habit [28,29]. These mental alterations can cause interpersonal problems because of changes in mood, behavior, or emotional stability. In this study, it was found that one of the most important cognitive changes is the increase of nervousness, which mainly occurs in men (65%), while in women its percentage is lower (38%). Another cognitive impairment most commonly reported is problems in remembering things or dates, both in men (30%) and in women (27%).

Cognitive alterations have been reported in several studies, where anxiety, fear, and nervousness are the most important mental alterations [29] and other studies in undergraduate students showed that after a post-infection by SARS-CoV-2 they manifested fear, nervousness, and stress [30]. This suggests that post-COVID-19 symptoms not only affect the pathophysiology of the individual but also alter the mental health of individuals, due to various factors such as quarantine [31].

Habit changes have also been related to the post-COVID-19 syndrome, such as alterations in eating and sleeping. In the present investigation, it was observed that mainly men (55%) changed their diet, while women did so to a lesser extent (27%). With respect to sleep habits, a similar percentage of both sexes, men (40%) and women (30%), reported this alteration (sleep disturbance). These changes may be due to social isolation, excessive use of mouth covers, as well as the quarantine affecting lifestyles and environment [32]. Anxiety and nervousness may be due to the risk of contagion and the social stigma of being infected [32]. Likewise, changes in habits could be due to the fact that during quarantine there was deregulation in their nap–wake-up schedule [33].

The variables that occur in post-COVID-19 symptoms are associated with different variables such as comorbidities (obesity, smoking, autoimmune diseases), sociodemographic factors (age, socioeconomic status, place of residence), or clinical situation (outpatient, hospitalized, or required intubation). An important point that has been associated with the persistence of post-COVID-19 symptoms is place of residence, since it has been observed that places with environmental contamination have repercussions, resulting in a slow recovery of the pulmonary damage (hypoxia) or also affecting the auditory contamination, which contributes to the appearance of headache or migraine. In this study, all the patients analyzed were residents of Tepic, Nayarit, which is an area with little air contamination. Since in this region air quality is considered satisfactory, this factor presents a minor or null risk. However, most of the participants mentioned that there was a lot of noise in their places of residence, so this could be a variable to consider as a risk for post-COVID-19 syndrome [34,35,36].

Other factors to consider include: socioeconomic status, since there are variables such as poor diet or an unhealthy lifestyle; insecurity; limited access to health systems, which can have repercussions in that the patient is not attended to in a timely manner once infected and therefore recovery from this infection takes longer than normal, causing persistent symptoms. Therefore, this is a relevant factor that should be evaluated in further research.

One of the concerns that has been generated in the world (mainly in Europe) is air pollution, since the increase of carbon monoxide or NO_2_ has repercussions in normal conditions on the induction of asthma, bronchitis, emphysema, and heart disease. As a result, in regions with high concentration of particles in the air, causes a higher prevalence of post-COVID-19 symptoms such as increased shortness of breath or hypoxia [37,38].

One of the most important variables is gender because it has been reported that women have a higher risk of post-COVID-19 symptoms, mainly depression, anxiety, stress, etc. In this study, we obtained results where men had a higher proportion of these symptoms, which may be due to various factors such as comorbidities or possibly the hormonal level or stress caused by the so-called home office [39].

A relevant symptom that was present mainly in men was fatigue. This is an interesting observation, since it is possible that the men surveyed presented a more severe clinical picture or associated comorbidities. Therefore, in future studies, it will be necessary to evaluate these associations of comorbidities and post-COVID-19 syndrome.

A factor related to the infection is the viral load since this has an impact on the severity of the disease [4] and also affects the number of symptoms during infection and post-COVID-19 [40,41]. In the present investigation, it was observed that patients who had a low Ct had between 15 and 20 symptoms, while people with a high Ct had fewer than six symptoms. One interesting finding is that asymptomatic patients [6] showed no or very few post-COVID-19 symptoms; this may be because they had a high Ct (~33–36), which would indicate the viral load measured by Ct. Therefore, patients with high viral load may manifest a higher number of symptoms and cognitive alterations post-COVID-19 and can be used as a reference to follow up with patients after SARS-CoV-2 infection.

Long-term and post-COVID-19 symptoms have been correlated, where the higher the viral load, the higher the risk of severe symptoms (respiratory problems, hyperventilation, low oxygen level, and even death); furthermore, it has been observed that this load also has an impact on post-COVID-19 symptoms [40,42,43]. In this study, we found a negative correlation between Ct value and post-COVID-19 symptoms, suggesting that there is a very high risk of post-COVID-19 syndrome with high SARS-CoV-2 infection.

Post-COVID-19 symptoms are of concern because people who recover from this infection may have disorders or symptomatology, which may cause family and work problems in the short- and long-term. Therefore, it should be a subject of study; it is also essential to know the time after which they can return to their activities. It has been reported that in other infections between 3 or 4 weeks of post-disease rest are needed, since there is a physical and mental imbalance [17,44], according to Koopmans et al., 2010. However, variations in prolonged symptomatology may be due to age, environment, vaccines, work, or pre-existing diseases of the person and should be taken into account for future research [44,45,46].

## 5. Conclusions

The predominant post-COVID-19 symptoms and cognitive alterations are fatigue and nervousness. These may be biased by sex and Ct (viral load) value, where a low Ct value may indicate that patients manifest a greater amount of physiological and mental alterations after SARS-CoV-2 infection.

**Limitations of this study**: One of the limitations of this research was that no patients in a chronic state (in the process of intubation) were evaluated, so only patients with mild or moderate symptomatology were included. Another limitation is that co-morbidities were not evaluated in the patients. A relevant limitation was the time of the survey since it could have been carried out over a longer period (6 months).

## Figures and Tables

**Figure 1 ijerph-19-15145-f001:**
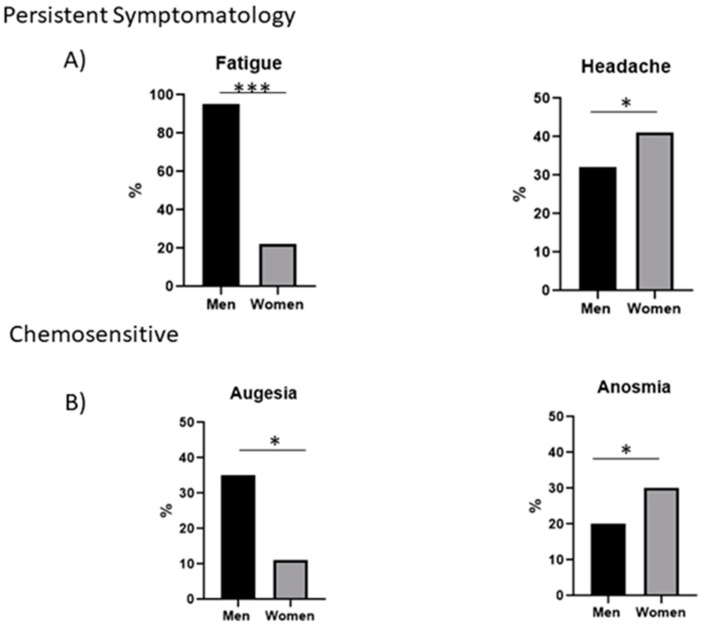
Persistent and chemo-sensitive symptoms post-COVID-19. (**A**) Percentage of men and women who present fatigue and headache. *n* = 76 (40 men and 36 women). The statistical test used was Student’s t-test. The statistical difference is *p* < 0.05 *, *p* < 0.0001 *** (**B**) Percentage of men and women with anosmia and augesia. *n* = 76 (40 men and 36 women). The statistical test used was Student’s t-test. The statistical difference is *p* < 0.05 *.

**Figure 2 ijerph-19-15145-f002:**
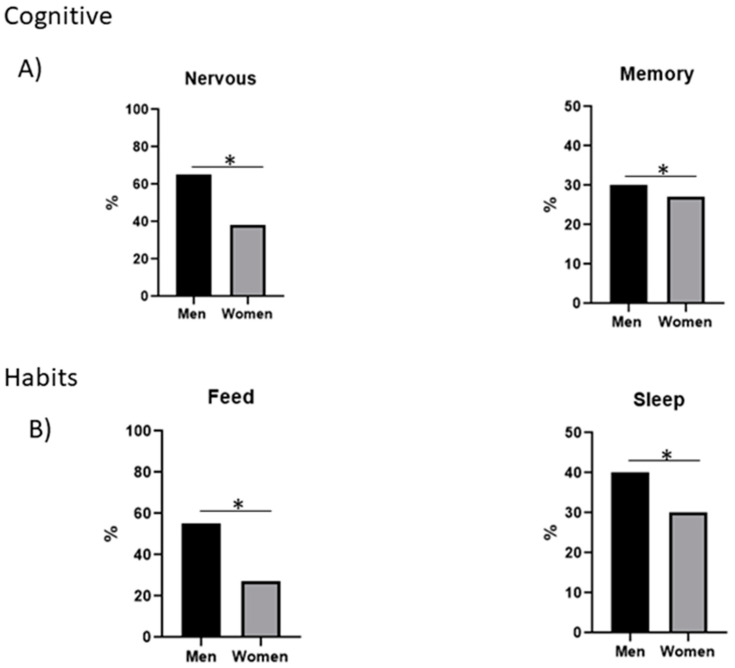
Cognitive symptoms and changes in habits post-COVID-19. (**A**) Percentage of men and women who present nervousness and memory loss. *n* = 76 (40 men and 36 women). The statistical test used was Student’s t-test. The statistical difference is *p* < 0.05 *, (**B**) Percentage of men and women who present change in diet and sleep. *n* = 76 (40 men and 36 women). The statistical test used was Student’s t-test. The statistical difference is *p* < 0.05 *.

**Figure 3 ijerph-19-15145-f003:**
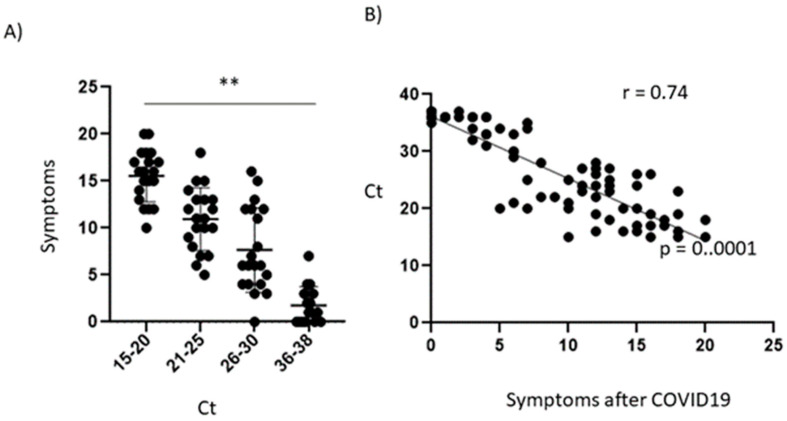
Association between the number of symptoms and Ct value and its correlation. (**A**) Number of symptoms present post-COVID19 versus the Ct value *n* = 76 (40 men and 36 women). The statistical test used was Student’s *t* test. The statistical difference is *p* < 0.001 **, (**B**) Pearson correlation (95% confidence level) and is reported as “r” and the probability value as “*p*”. *n* = 76.

**Table 1 ijerph-19-15145-t001:** Percentage of post-COVID-19 signs and symptoms according to sex.

	Total	Male	Females
Patients	76	40	36
Age (years)	20–70	45 ± 25	30 ± 25
Patients with no symptoms after COVID-19 infection	6	5	1
Persistent clinical symptomatology			
Fatigue	46	38	8
Muscle pain	22	13	9
Headache	28	13	15
Cough	15	10	5
Diarrhea	12	7	5
Nausea	2	0	2
Chemosensitive problems			
Smell	19	8	11
Taste	18	14	4
Vision	10	6	4
Audition	2	1	1
Cognitive skill changes			
Nervousness	40	26	14
Remembering problems	22	12	10
Confusion	19	11	8
Happiness	10	7	3
Sadness	16	8	8
Sudden discomfort	10	6	4
Mood	10	6	4
Fear	16	9	7
The feeling of change (partner, house)	11	7	4
Decision-making problems	7	6	1
Problems in expressing oneself	4	4	0
Mathematical troubleshooting problems.	8	6	2
Impulsive	4	4	0
Habit changes			
Feeding	32	22	10
Sleep	27	16	11

**Table 2 ijerph-19-15145-t002:** Value of cycle threshold (Ct) and relationship with the viral load of SARS-CoV-2 based in Gene E.

Load Viral of SARS-CoV-2 Based on E Gene	
Ct	Log Copy/µL
20<	<451,743
20	451,743
21	241,631
22	129,245
23	69,131
24	36,977
25	19,779
26	10,579
27	5659
28	3027
29	1619
30	866
31	463
32	248
33	133
34	71
35	38
36	20
37	11
38	6
39	3
40	2

## Data Availability

All data supporting the findings of the study are available from its corresponding author, G.-P.M.I., upon reasonable request.

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
