# Peer review of "Post-COVID-19 Syndrome in Outpatients and Its Association with Viral Load"

_ijerph, 2022, doi:10.3390/ijerph192215145_

Round 1

Reviewer 1 Report

First of all, congratulations to the authors for their study on post-covid 19 syndrome. Once the pandemic is 'under control', one of the main concerns of science should be to analyze the mid- and long-term consequences of having been infected. One of the strengths of the work are the identification of different degrees of relations between CT and post-covid symptoms. Having said the above, we believe that the text would be improved if the authors would heed the following advice:

-Edit words in red

-Compare the results of the study with others carried out elsewhere in the world, and from this analyze the particularities -for example, how particular sociodemographic variables may have an impact on these results- and those issues that are generalizable regardless of the specific context in which the research has been carried out.

-Similarly, it would be interesting for the authors to adopt a less descriptive and more interpretative tone when analyzing the impact of variables such as sex or age on post-covid syndrome according to the established categories.

Best regards

Author Response

Dear Editor, we deeply appreciate the revisions made by the reviewers, we consider that they have enriched and strengthened our work. Therefore, we have reviewed and addressed the comments and suggestions given, below you will find the responses to the reviewers' comments.

Response for reviewer 1

Reviewer.- Compare the results of the study with others carried out elsewhere in the world, and from this analyze the particularities -for example, how particular sociodemographic variables may have an impact on these results- and those issues that are generalizable regardless of the specific context in which the research has been carried out.

Authors.- Thanks for your observations, we added the new sentences in the discussion

The variables that occur in post-COVID-19 symptoms are associated with different variables such as comorbidities (obesity, smoking, autoimmune diseases), sociodemographic (age, socioeconomic status, place of residence), or clinical situation (outpatient, hospitalized, or required intubation).

An important point that has been associated with the persistence of post-COVID-19 symptoms is the place of residence since it has been observed that places with environmental pollution have repercussions in a slow recovery of pulmonary damage (hypoxia) or also affects auditory pollution, since this contributes to the appearance of headache or migraine, in this study all the patients analyzed were residents of Tepic, Nayarit, for which this is an area with little air pollution, since in this region it is considered satisfactory, it presents a minor or null risk; However, most of the participants mentioned that there was a lot of noise in their places of residence, so this could be a variable to consider as a risk for the post-COVID-19 syndrome.

Other factors to consider include socioeconomic status, since there are variables such as poor diet or an unhealthy lifestyle, insecurity, and limited access to health systems, which can have repercussions in that the patient is not attended to in a timely manner once infected and therefore recovery from this infection takes longer than normal, causing persistent symptoms. Therefore, this is a relevant factor that should be evaluated in further research.

One of the concerns that have been generated in the world (mainly in Europe) is air pollution, since the increase of carbon monoxide or NO2 has repercussions in normal conditions in the induction of asthma, bronchitis, emphysema, and heart disease, so that, in regions with a high concentration of particles in the air, causes a higher prevalence of post-COVID-19 symptoms such as increased shortness of breath or hypoxia. 

Similarly, it would be interesting for the authors to adopt a less descriptive and more interpretative tone when analyzing the impact of variables such as sex or age on post-COVID-19 syndrome according to the established categories.

Thanks for your observations, we added the new sentences in the discussion

One of the most important variables is gender because it has been reported that women have a higher risk of post-COVID-19 symptoms, mainly depression, anxiety, stress, etc. In this study we obtained results where men had a higher proportion of these symptoms, this may be due to various factors such as comorbidities or possibly the hormonal level or stress caused by the so-called home office.

A relevant symptom that was presented mainly in men was fatigue in men, this is interesting data, since it is possible that the men surveyed presented a more severe clinical picture or associated comorbidities, so in future studies, it is necessary to evaluate these associations of comorbidities and post covid-19 syndrome.

The English was checked

Reviewer 2 Report

I consider it is a very important work, however limited to mild or asymptomatic cases, although it opens the possibility of studying the same variables in patients who were with severe COVID symptoms or in a critically ill condition, in whom the post-COVID syndrome will surely be more incident, having a directly proportional relationship of the persistence of symptoms according to the severity of the condition.

Regarding some symptoms that can act as a good or bad prognosis or null or great severity, there is a small Mexican study that deals with it and I would recommend you to analyze

https://www.ncbi.nlm.nih.gov/pmc/articles/PMC8498418/

Author Response

Response to reviewer 2

Reviewer 2.- I consider it is a very important work, however limited to mild or asymptomatic cases, although it opens the possibility of studying the same variables in patients who were with severe COVID symptoms or in a critically ill condition, in whom the post-COVID syndrome will surely be more incident, having a directly proportional relationship of the persistence of symptoms according to the severity of the condition.

Regarding some symptoms that can act as a good or bad prognosis or null or great severity, there is a small Mexican study that deals with it and I would recommend you to analyze

Authors.- Thanks for your recommendation we read the paper and added some sentences to the manuscript.

COVID-19 infection has been reported to cause long-term neurological alterations, such as cerebrovascular diseases or inflammatory myopathies type 1. Other symptoms, such as anosmia and ageusia, may manifest after the end of the infection and may be used as prognostic markers for neuronal diseases. Other manifestations associated with this disease are the increase in migraine or cases of epilepsy; however, the evidence is still inaccurate because it may be due to the infection or the mandatory confinement.

Some sentences of the manuscript recommended were added.

The English was checked

Reviewer 3 Report

Dear authors,

The research topic is valuable in its scope and content, but I would like some major corrections. I hope that after the necessary corrections, your research will be able to be published in the journal Ä°JREPH. The relevant fixes are below.

Abstract

-Please elaborate this section by dividing it into titles (Background and objectives, method, results and conclusion). In particular, please detail the method part (which tests and surveys were used)

Introduction

Please include a few more examples of research done in the introduction and reveal the original value of your research.

Also, please include the main hypothesis at the end of the purpose sentence.

I think the method and results section is sufficient.

Discussion

I liked your discussion, but there is no information about the limitations of your research. Please add the limitations of your research as a new paragraph at the end of this section.

Yours sincerely

Author Response

Response to reviewer 3

Reviewer 3.- The research topic is valuable in its scope and content, but I would like some major corrections. I hope that after the necessary corrections, your research will be able to be published in the journal Ä°JREPH. The relevant fixes are below.

Reviewer 3.- Abstract-Please elaborate this section by dividing it into titles (Background and objectives, method, results and conclusion). In particular, please detail the method part (which tests and surveys were used)

Authors.- Thank you for your recommendation, we have added a new abstract.

Reviewer 3.- Introduction Please include a few more examples of research done in the introduction and reveal the original value of your research. Also, please include the main hypothesis at the end of the purpose sentence. 

Authors.- We added new sentences in the introduction and re-wrote some parts.

The Post-COVID-19 syndrome has had different pathologies associated with pulmonary (hypoxia, lung damage or fibrosis) hematologic (thromboembolic events, longer duration of inflammatory events) cardiovascular (dyspnea and prolonged chest pain) neurological (sleep disruption, depression, fatigue) renal (kidney damage) endocrine (bone demineralization, uncontrolled diabetes, thyroiditis) dermatologic (hair loss) systems. All these conditions should be approached from a multidisciplinary point of view and identify variables that can predict the susceptibility of these conditions. One of these variables is the viral load, which can be defined as Load viral is defined by the amount of a virus in an organism, (bloodstream or tissues) typically quantified by virus particles per milliliter.

The relationship between SARS-CoV-2 virus detection at different time points and its viral load may aid in the clinical interpretation of post-COVID-19 syndrome. In addition, it has been described that the virus can survive longer in feces compared to respiratory tract secretions. This may result in viral proteins that may not cause infection, but trigger and exacerbate a prolonged immune response, which could explain the presiding symptoms associated with COVID-19 .

There are few studies regarding whether the viral load determined by Ct, has any impact on the different post-COVID-19 effects, either in persistent symptomatology or that these alterations are due to risk factors associated with the individual environment of patients [11]. Therefore, this paper aimed to evaluate the post-COVID-19 symptoms prevalent in patients and to analyze the correlation between Ct and these symptoms after infection.

Reviewer 3.- I think the method and results section is sufficient.

Discussion 

Reviewer 3.- I liked your discussion, but there is no information about the limitations of your research. Please add the limitations of your research as a new paragraph at the end of this section.

Authors.- Thanks for your recommendation we added a new section study limitation

Limitations of this study: One of the limitations of this research was that patients in a chronic state (in the process of intubation) were not evaluated, so only patients with mild or moderate symptomatology were evaluated. Another limitation is that co-morbidities were not evaluated in the patients. A relevant limitation was the time of the survey since it could have been carried out over a longer period (6 months).

The English was checked

Reviewer 4 Report

Dear Editors/authors, 

Have a nice day. 

This paper studies 76 COVID-19-positive patients to identify an association between Post-COVID-19 syndrome and viral load. The study is interesting and catches the attention of readers. Here are some minor suggestions to improve the quality of this work. 

TITLE: The title is fine. 

ABSTRACT:

1. How many patients are sampled and the methods for this study should be informed in one line. 

INTRODUCTION:

2. Line 52 , what does it mean by CITA?  

3. Similar to "post-COVID-19 syndrome" defined in line 59, a short description of "viral load" should be added. 

METHODS

4. Methods adopted are not rigorous but serve the cause.

RESULTS

5. The results provided are in line with the research questions and are clearly explained.  

Author Response

Response to reviewer 4

Have a nice day. 

Reviewer 4. This paper studies 76 COVID-19-positive patients to identify an association between Post-COVID-19 syndrome and viral load. The study is interesting and catches the attention of readers. Here are some minor suggestions to improve the quality of this work. 

Reviewer 4. TITLE: The title is fine. 

Reviewer 4.- ABSTRACT: 1. How many patients are sampled and the methods for this study should be informed in one line. 

Authors.- We added in the abstract this information

Introduction The COVID-19 pandemic is caused by the SARS-CoV-2 virus, which has caused more than 100 million infections and more than 2.5 million deaths worldwide, representing a serious public health problem. The gold method to detect this virus is qRT-PCR which is a semiquantitative technique, where the viral load can be known through its cycle threshold (Ct). It has also been reported that COVID-19, generates long-term symptoms (post-COVID-19). Methods after three months a survey was performed to 70 COVID-19 confirmed patients, after that we divided into 4 groups (symptoms persistent, chemo-sensitive, cognitive, and habit changes), to determine the correlation between viral load and post-COVID-19 symptoms. Results Data show that fatigue, nervousness, anosmia, and diet changes, are the common long-term symptoms; in addition, a negative correlation was found between viral load and the number of post-COVID-19 symptoms. Conclusion. The COVID-19 generates long-term symptoms and these can generate problems that can have psychological and social repercussions.

Reviewer 4.- INTRODUCTION: 2. Line 52 , what does it mean by CITA?  

Authors.- Sorry, this was a mistake, it was fixed.

Reviewer 4.- 3. Similar to "post-COVID-19 syndrome" defined in line 59, a short description of "viral load" should be added. 

Authors.- The meanings of viral load were added.

Load viral is defined by the amount of a virus in an organism, (bloodstream or tissues) typically quantified by virus particles per milliliter.

Reviewer 4. METHODS

  1. Methods adopted are not rigorous but serve the cause.

Reviewer 4. RESULTS

  1. The results provided are in line with the research questions and are clearly explained.  

 The English was checked

Round 2

Reviewer 3 Report

Thank you for your efforts. congratulations